# The Geriatric Depression Scale Predicts Glycemic Control in Older Adult with Type 2 Diabetes Mellitus: A Longitudinal Study

**DOI:** 10.3390/healthcare10101990

**Published:** 2022-10-11

**Authors:** Thanitha Sirirak, Pasuree Sangsupawanich, Nahathai Wongpakaran, Wisarut Srisintorn

**Affiliations:** 1Department of Family and Preventive Medicine, Faculty of Medicine, Prince of Songkla University, Songkhla 90110, Thailand; 2Department of Pediatrics, Faculty of Medicine, Prince of Songkla University, Songkhla 90110, Thailand; 3Department of Psychiatry, Faculty of Medicine, Chiang Mai University, Chiang Mai 50200, Thailand

**Keywords:** type 2 diabetes mellitus, depression, older adult, type 2 diabetic older patients, geriatric depression scale, longitudinal study

## Abstract

The presence of comorbid depression and diabetes is associated with worse glycemic control, higher complication and greater mortality risk than expected by each condition alone. The association between various levels of severity of depressive symptoms and glycemic control over time among type 2 diabetic older patients was unclear. This study aimed to investigate a longitudinal association between depression and HbA1c among type 2 diabetic older patients. Type 2 diabetes patients aged 60 years and above with normal cognition were recruited from the outpatient department from 1 June 2020 to 1 July 2021. The Thai Geriatric Depression Scale (TGDS) and HbA1c were assessed at five time points (baseline and every 12 weeks) for 1 year. A linear mixed effect model was used. Of the 161 enrolled participants, 146 completed the study. At baseline, 14% were susceptible to depression or having depression (TGDS score 6 and above), and there was a significant correlation between HbA1c and depression (r = 0.26, *p* ≤ 0.01). The longitudinal analysis indicated that TGDS was a significant predictor of HbA1c in the next visit, and the relationship was J-shaped. A TGDS below 5 was associated with decreasing HbA1c in the next visit, but the association became positive at a TGDS score at 5 or higher. The presence of significant symptoms of depression was associated with glycemic control in the next 3-month interval OPD visit event, although major depressive disorder has not yet been established.

## 1. Introduction

Diabetes mellitus (DM) is the chronic illness that impacts on people’s health in all dimensions—physically, psychologically and socially. Over the 30-year period, DM is one of the top three conditions (HIV/AIDS and other musculoskeletal disorders) that are contributing to the large increases in age-standardized disability-adjusted life-years (DALYs) rates between 1990 and 2019, with an increase of 24.4% [1]. DM is currently ranked as the 8th and the 3rd leading cause of global disease burden for all age groups and for age 50–74 years, respectively [1]. Depression is one of the ten leading causes that had the largest absolute increases in number of DALYs and is also common in all age groups from adolescence into older age [1]. With the increasing age, DM and depression are among the most prevalent chronic illnesses worldwide. According to longitudinal study, people with DM are two to three times more likely to be diagnosed with depression compared to those without DM [2]. The most recent real-world settings study called the International Diabetes Management Practices Study observed moderate depressive symptoms in 8–16% of patients with type 1 or type 2 DM [3]. A meta-analysis of 26 cross-sectional studies revealed a significant association of depression with hyperglycemia [4], and data from systematic reviews showed that the presence of depression in diabetes led to a worsening of the metabolic and HbA1c, an increased risk of microvascular and macrovascular complications, deterioration in the quality of life and risk of more severe course of depression [2]. Gathering evidence from cohort and systematic reviews with meta-analysis studies, depression was significantly associated with an almost 1.25 to 2.24-fold increased risk of all-cause mortality in people with DM [5,6,7,8].

Data from the Thai National Health Examination Survey 2004–2014 have shown that the prevalence of DM significantly increased with age and peaked at 60 to 69 years old [9]; meanwhile, normal aging is associated with a progressive increase in HbA1C [10]. Specifically, depression affects approximately 17 to 80% of the Thai older adults in the community depending on the setting and measurement [11,12,13]. 

Of the few longitudinal studies, only one by Richardson et al. studied type 2 older diabetic patients and demonstrated that depression is associated with persistently higher HbA1c levels [14]. These findings were consistent with other longitudinal studies by Lustman et al. and Ching-Ju Chiu et al. in middle-aged and older type 2 diabetic patients [4,15]. In contrast, Aikens et al. and Fisher et al. indicated that depression did not predict any change in HbA1c levels in prospective analysis [16,17]. 

Older adults are focused on the treatment of medical conditions and do not adequately recognize the symptoms of depression, whose symptom are less clear cut in a primary care setting than those seen in a psychiatry clinic. Prior studies of the association between depression and glycemic control were mostly cross-sectional in nature. Meanwhile, previous longitudinal studies provided insufficient information of depression as a predictor of glycemic control among older adults in a primary care setting. Therefore, the main purpose of this study is to demonstrate the association between depression and glycemic control in type 2 diabetic older patients. Based upon available evidence in the literature, the hypothesis is that depression would be associated with higher HbA1c levels in the next visit (HbA1c next visit) in the type 2 diabetic older patients. This association will provide the proper care plan of depression screening in a primary care setting. 

## 2. Materials and Methods

### 2.1. Study Design and Participants

This is a longitudinal study that aimed to analyze the relationship of depression with subsequent glycemic control over a one-year follow-up period in which the variables were repeated measure at 3-month duration 5 times. Participants were recruited from older patients (age 60 years and above) diagnosed with type 2 diabetes mellitus and ongoing treatment at the outpatient department (OPD) of a primary care setting that provided service by university-affiliated physicians in Southern Thailand. The type 2 diabetic older patients who came for routine visit during the period of study on 1 June 2020 to 1 July 2021 received the invitation poster that was approved by the ethical committee and voluntarily applied for participation. 

Participants with an ICD-10 code for dementia, cognitive impairment, depressive disorder or anxiety, bipolar disorder, psychotic disorder, alcoholism or drug addiction within 1 year prior to the first visit or malignancy within 5 years prior to the first visit were excluded. Cognitive function and risk of suicide were evaluated at baseline using the Thai Mini-Mental State Examination (MMSE-Thai 2002) and 8-questions assessment of suicide risk tool (8Q), respectively, to exclude participants with cognitive impairment and active suicidal ideation from the study. The high suicidal risk subjects (8Q score more than 17) were sent to a psychiatric clinic.

Among 161 participants, 146 completed the 5 visits (Time 0 to Time 4) during one year of longitudinal study. Eight participants who declined hospital visits due to the COVID-19 pandemic withdrew from the study, and 7 participants were lost to follow-up. The majority of the participants were independent and socially active; meanwhile, 8 participants (5.5%) needed assistance in answering the questionnaires.

### 2.2. Study Instruments

We recorded the demographic characteristics (age, gender); type 2 diabetes-related variables including years of having DM, medications for type 2 DM (no insulin vs. using insulin), presence of diabetic retinopathy and presence of diabetic nephropathy; and cardiovascular risk factors consisted of body mass index (BMI), cigarette smoking and low-density lipoprotein (LDL) cholesterol. 

Level of physical activity in the past seven days was categorized as low, moderate and vigorous intensity using the short form of The International Physical Activity Questionnaires (IPAQ)-Thai version which appeared to have acceptable validity and reliability (Spearman’s correlation 0.22 and intra-class correlation coefficient 0.69) [18]. The IPAQ short form asks about three specific intensity of activities undertaken in domains including leisure time, domestic and gardening activities, work-related and transport-related activity;

Nutritional status was assessed using the Mini Nutritional Assessment (MNA) scale in Thai version. MNS is a screening for undernutrition in geriatric practice [19]. Participants were interviewed by the examiners; the score 12 to 14 indicates “normal nutritional status”, 8 to 11 indicates “at risk of malnutrition”, and less than 8 indicates “malnourished”. 

Quality of life was determined using the 5-level EQ-5D (EQ-5D-5L) Thai version [20] which consists of 2 parts, the EQ-5D descriptive system and the EQ visual analogue scale (EQ VAS). The EQ VAS records the subject’s self-rated health status, where the endpoints of the visual analogue scale are labeled ‘The best health you can imagine’ and ‘The worst health you can imagine’. The score of 100 indicates the best health status.

Depressive symptoms were assessed by the Thai Geriatric Depression Scale −15 items (TGDS). It was translated by Wongpakaran et al. in 2012 and found to perform well at the geriatric outpatients care with the cut-off score of >5, a sensitivity of 0.92 and a specificity of 0.87 [21]. The Geriatric Depression Scale focuses specifically on psychiatric rather than somatic symptoms (sleep disturbances, weight loss and pessimism about the future), which can be related to aging rather than the experiences of younger adults. The questionnaire can be self-administered or presented as an interview. Most of the participants in this study answered the 15 questions in the yes/no format themselves. For the older adults with eye problems and low literacy, the trained examiner read all items to them in a clear monotone voice without facial expression. Each item scored one point if positively answered except for items 1, 5, 7, 11, and 15, which received points when negatively answered. Sum scores of 0–5 are considered normal, 6–10 indicated susceptible to depression, and 11–15 indicated that the patient has depression. 

Hemoglobin A1C (HbA1C), serum total cholesterol, high-density lipoprotein (HDL) cholesterol, triglycerides and LDL cholesterol were measured at every visit. Blood samples were drawn after an overnight fast (at least 8 h fasting). The capillary electrophoresis method was performed using the ethylene diaminetetracetate (EDTA) blood for evaluated HbA1c. The direct homogenous (automate) method was performed using lithium heparinized plasma for evaluating LDL and HDL cholesterol.

Thereafter, participants received routine diabetes care in OPD, which was provided by a physician following detailed treatment algorithms based on “Treatment of Diabetes in Older Adults: An Endocrine Society Clinical Practice Guideline” [22] to maintain their established oral hypoglycemic and/or insulin therapy.

### 2.3. Sample Size Estimation

The sample size calculations for longitudinal data were completed using longpower package, R program [23]. The authors assume a weak correlation coefficient (rho) in the range of −0.3 < rho < 0.3 and the large error variance (sigma2). The sample size when rho = 0.2 and sigma2 = 300 is 132.

### 2.4. Statistical Analysis

Descriptive results of the participants in this study are expressed as the calculation of means and standard deviation (SD) for continuous variables and as frequencies and percentages for categorical variables. This longitudinal study aimed to analyze the relationship of depression with subsequent HbA1c in the next visit in which the variables were repeatedly measured at 12-week durations 5 times. 

To perform prospective analyses, a linear mixed effect model with random effects to adjust for variance within individuals and variation between individuals over time were implemented. There were two levels of variance components. Level 1 considered within-person change over time, and level 2 considered between individuals’ variation, regardless of time. Time-variant covariates including HbA1c, TGDS, BMI, diabetic complications, medication, nutritional status, cigarette smoking, quality of life and physical activity were collected at each time point from T0 to T4 in a 12-week duration. As HbA1c is the test that reflects the average plasma glucose of the 12 weeks previously, we consider that the individual differences on current HbA1c measured (time *n*) are related to individual differences on other variables measured at a previous occasion (time *n*-1) as shown in Figure 1.

Data were analyzed using the R program with 2-sided tests of hypotheses. A *p*-value < 0.05 on the two-tail was considered to indicate statistical significance.

### 2.5. Ethics

This study was conducted in accordance with the ethical principles described in the Helsinki declaration. Written informed consent was obtained from all participants. Participants were voluntary, and they could refuse to participate or withdraw from the study at any time. The study was approved by the Ethical Committee of Faculty of Medicine, Prince of Songkla University (REC 62-078-9-1). 

## 3. Results

### 3.1. Baseline Assessments

Of the 146 participants who completed five visits (T0 to T4) during one year of longitudinal study, the majority were female (61.9%) and married (78.2%). The mean age was 67.9 years, with a standard deviation of ±5.8 years. The average (SD) duration of diabetes was 10.9 (5.9) years. The mean (SD) HbA1c was 7.5% (1.3), ranging from 5.4 to 12%. Diabetic complications were common among the participants; these included diabetic retinopathy (DR) (66.7%) and diabetic nephropathy (DN) (44.2%). In addition to an oral hypoglycemic, 21.9% were prescribed insulin. Twenty eight percent of participants had been using more than five medications, which indicated polypharmacy. Seventeen participants (11.6%) were susceptible to depression, and three participants (2.7%) were considered depressed at baseline. Baseline demographic and clinical characteristics are described in Table 1.

At baseline, HbA1c had a significant correlations with TGDS (r = 0.26, *p =* 0.001, 95%CI 0.11 to 0.41), duration of DM (r = 0.24, *p =* 0.003, 95%CI 0.09 to 0.39), MNA score (r = −0.24, *p =* 0.003, 95%CI −0.39 to −0.08) and using insulin. The mean of HbA1c was significantly higher among participants using insulin (*p* < 0.001, 95%CI −1.68 to −0.73), as shown in Figure 2. 

A line graph shown in Figure 3 represents how HbA1c and TGDS have changed over the same period of time at 3-month intervals. It can be seen that the mean and 95% CI of HbA1c and TGDS remain steady over a 1-year period.

### 3.2. Prospective Analysis of Depression as a Predictor to Glycemic Control

In this analysis, we entered a quadratic function for TGDS (of the form ax^2^ + bx + c) to models to allow for a non-linear relationship. This model demonstrated the association between TGDS and HbA1c, adjusted for age, gender, duration of DM, cigarette smoke, BMI, level of physical activity, nutritional status, insulin used, DN, DR, LDL cholesterol and EQ VAS (Table 2). When other predictors were fixed, an increase of 1 point of TGDS among 0 to 5 corresponds to a decrease in HbA1c. Meanwhile, an increase of 1 point of TGDS among 5 to 13 corresponds to an increase in HbA1c. This analysis indicated that TGDS was a significant predictor of HbA1c. The relationship between TGDS and HbA1c was J-shaped (Figure 4). The shaded area is a 95% confidence interval for the fitted values, which was based on standard errors computed from the covariance matrix of the fitted regression coefficients. The rug plot at the bottom of Figure 1 determines the location of TGDS. 

## 4. Discussion

In this baseline analysis, 14% of participants had a TGDS score of 6 and above, which indicated that the baseline TGDS in this study is close to the prevalence of depression among the older adults in the community studied by Thongtang et al. [12] and in the general outpatient clinic studied by Anantapong et al. [24], which were 12.7% and 9.6%, respectively. 

The results of this study demonstrate that a TGSD score of 6 and above is associated with the subsequent increase in HbA1c over 1 year follow-up after health-related characteristics that may be associated with HbA1c had been adjusted. Prior systematic reviews with a meta-analysis of six longitudinal studies demonstrated that the presence of depression showed a small but significant increased risk of having higher HbA1c, and it was also a bidirectional association [25]. Even though the participants in this study were older adults, the result was still consistent with prior longitudinal studies that suggested depression is associated with higher HbA1c levels over time [14,15,16,25]. The link between depression and consequently hyperglycemia could be explained by self-management behaviors and self-efficacy. Higher depressive symptoms have been associated with fewer self-management behaviors and low self-efficacy, which has in turn been related to hyperglycemia [26]. 

GDS is one of the screening tools that has been used in many studies and has been translated and validated into many languages, including the Thai version called TGDS by Wongpakaran et al. [21]. To our knowledge, this is the first study that reports a non-linear “J-shaped” relationship of GDS with HbA1c.

Similar associations were observed for the other depressive screening tools. The previous longitudinal study by Abuhegzy et al. in 2017 found a significant association using the Beck Depression Inventory score with hyperglycemia [27]. A study of the association between cardiovascular risk factors and concurrent depressive symptoms using the hospital anxiety and depression score (HADS-D) in patients with three common cardiometabolic conditions by Bhautesh et al. in 2014 demonstrated the J-shape association of HbA1C with the probability of having a positive HADS-D (>7). Not only HbA1c but systolic blood pressure, diastolic blood pressure, BMI and total cholesterol also demonstrated the J-shape association with probability of having a positive HADS-D [27]. Studies also demonstrated a non-linear “J-shaped” relationship between GDS and various health variables. The Health in Men Study found the “J-shaped” association between the numbers of years lived with the diagnosis of DM and an increase in the odds of current depression [28].

Interestingly, a TGDS score of less than 6 had negative association with HbA1c. The explanation should be one of the limitations of our study: we did not evaluate other mental health conditions that can alter HbA1c or the relationship between depression and HbA1c, which include stress, other mood disorders such as bipolar disorder, and diabetes distress. Stress that lasts for many weeks or months can lead to unstable levels of blood glucose. The fluctuations of blood glucose, whether high or low, may cause a person to experience a variety of symptoms such as hunger, confusion, aggression and irritability. Stress is a normal part of daily life; people with stress can live without depression, but it may lead to unhealthy eating behaviors that affect glycemic control. It was noticeable that older adults without depression had HbA1c scores less than 7.65 (Figure 4). Although there was a negative correlation between a TGDS score of less than 6 and HbA1c, the level of HbA1c was still close to the HbA1c target levels recommended by the American Diabetic Association (ADA) position statement “Older Adults: Standards of Medical Care in Diabetes—2022”, that healthy older adults with intact cognitive function and functional status should have an HbA1c goal of less than 7.0–7.5% [29]. 

The importance of identifying depression among type 2 diabetic older patients cannot be overstated. Aside from affecting the patient’s psychosocial life, depression has been linked to poor glycemic control, poor adherence to medication and dietary regimens, and overall reduction in quality of life [7,30]. Thus, the ADA as of 2022 screens for geriatric syndromes (i.e., polypharmacy, cognitive impairment, depression, urinary incontinence, falls, persistent pain, and frailty) in older adults, as they may affect diabetes self-management and diminish quality of life (B-level evidence) [29]. Furthermore, psychosocial assessment, treatment and referral to a Mental Health Specialist is preferable into routine care rather than waiting for a specific deterioration in physiological or psychological illness [31].

The goal of this study is to demonstrate the association between TGDS, a simple self-reported depressive symptoms questionnaire, and the next HbA1c visit. This result highlights the importance of comprehensive care in the type 2 diabetic older patients to not only focus at target HbA1c levels or DM complication but also consider screening for depression as a part of DM care. The older adults with depressive symptoms or disorder need ongoing monitoring of depression recurrence within the context of routine care. 

Our study had several strengths; these include the prospective design, which can provide better quality of data on HbA1c and depression, because the HbA1c level was obtained from the participant at the same time as depression was identified by the TGDS. Furthermore, the prospective repeated measured data of comorbidity and other clinical and confounding factors allow the detection of within-person change over time. Another strength is that this study is the first to evaluate the association of depression as a predictor of HbA1c specifically in the type 2 diabetic older patients. 

This study has some limitations. The first limitation is the use of screening instrument: TGDS is not a diagnostic test for depressive disorder, so it is unknown whether our findings are fully applicable to the older adults with clinical depressive disorder. The second limitation is that enrollment occurred within one primary care center, which may introduce selection bias. The characteristics of type 2 diabetic older patients in this study include 10 years on average of DM duration, polypharmacy, many comorbid diseases, the presence of microvascular complication, and insulin usage, which represent the diabetic patients with chronic hyperglycemia and more severe underlying illnesses. The findings in this study may not apply to the patients with a new diagnosis with DM. The last limitation that we will mention is that only 2.1% reported certainly having depression (TGDS score 11–15), so the correlation between severe depression and glycemic control was not apparently identified.

In closing, the finding indicates that there is the relationship between depression and a subsequent increase in HbA1c in DM patients age 60 and above. The relationship was observed only when TGDS is used to measure susceptibility for depression (6 and above). 

## 5. Conclusions

This result meets our hypothesis that depression is an important factor in glycemic control in the type 2 diabetic older patients. Our result provides crucial information regarding an association between the presence of significant symptoms of depression and glycemic control in the next 3-month interval OPD visit event, although major depressive disorder has not yet been established. So, mental health evaluation is important to routinely screen in type 2 diabetic older patients in every OPD visit, and a treatment plant or preventive program should be provided as soon as possible. 

## Figures and Tables

**Figure 1 healthcare-10-01990-f001:**
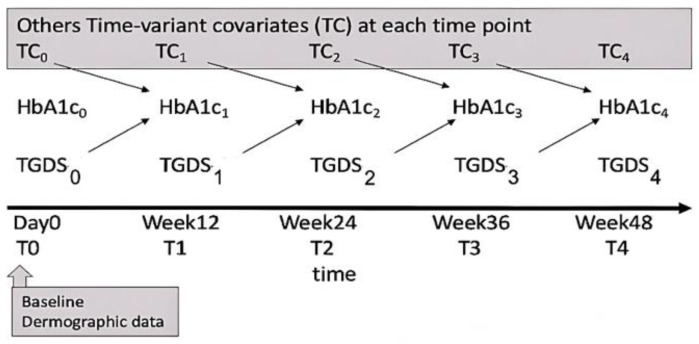
Schematic of data collection demonstrates that from time 0 to time 4, the variables were repeated measure at 12-week duration 5 times. The individual differences of TGDS and TC at each occasion from time 0 to time 3 were related to individual differences of HbA1c in the latter occasion or the next visit.

**Figure 2 healthcare-10-01990-f002:**
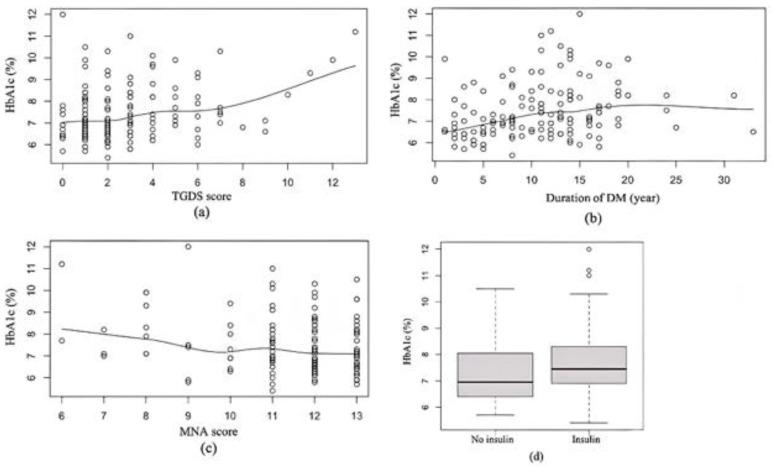
Baseline correlations between HbA1c and (**a**) TGDS score, (**b**) duration of DM, (**c**) Mini Nutritional Assessment (MNA) score and (**d**) insulin (no insulin vs. insulin).

**Figure 3 healthcare-10-01990-f003:**
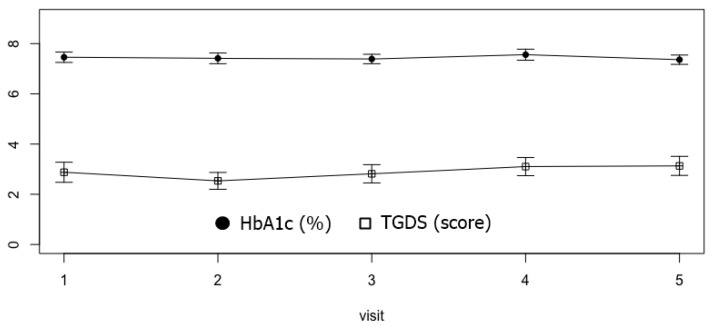
Mean and 95% confidence interval at each time point from visit 1 (T0) to visit 5 of HbA1c and TGDS.

**Figure 4 healthcare-10-01990-f004:**
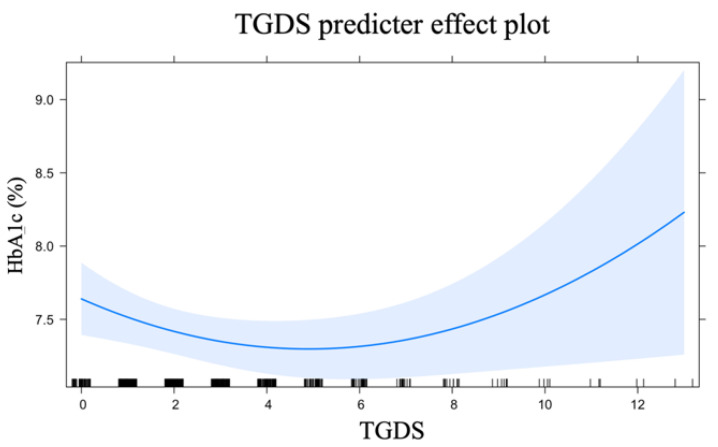
The predictor effect plots of TGDS as a predictor to glycemic control.

**Table 1 healthcare-10-01990-t001:** Baseline demographic data and clinical characteristic of participants at first visit (T0).

Baseline Characteristics		*n* (%)
Gender	Female	91 (61.9)
	Male	55 (38.1)
Marital status	Couple	115 (78.8)
	Widow	32 (21.8)
Cigarette smoker	Never	103 (70.1)
	Former	34 (23.1)
	Current	10 (6.8)
Physical activity (intensity)	Low	42 (28.6)
	Moderate	67 (45.6)
	Vigorous	38 (25.9)
Nutritional status	Normal	88 (59.9)
	Risk of malnutrition	59 (40.1)
TGDS	No depression (<6)	126 (85.7)
	Susceptibility (6–10)	17 (11.6)
	Depression (11–15)	3 (2.7)
Present of diabetic complication	Diabetic retinopathy	98 (66.7)
	Diabetic Nephropathy	65 (44.2)
	Diabetic foot	85 (57.8)
Diabetic Medication	No	5 (3.4)
	Oral hypoglycemic agent only	109 (74.7)
	Oral hypoglycemic agent + insulin	26 (17.8)
	Insulin only	6 (4.1)
Comorbid disease	Hypertension	118 (80.3)
	Dyslipidemia	137 (93.2)
	Cardiovascular disease	14 (9.5)
	Musculoskeletal disease	64 (43.5)
	Respiratory disease	10 (6.8)
		**mean (SD)**
Age (year)		67.9 (5.8)
Duration of DM (year)		10.9 (5.9)
EQ VAS		75.1 (17.2)
HbA1c (%)		7.45 (1.3)
Fasting plasma glucose (mg/dL)		143 (90.8)
LDL cholesterol (mg/dL)		109 (36.4)
HDL cholesterol (mg/dL)		53.1 (14.4)
Triglyceride (mg/dL)		133 (60.8)
Body mass index (kg/m^2^)		26.8 (4.8)
Systolic blood pressure (mmHg)		138 (13.7)
Diastolic blood pressure (mmHg)		73.5 (9.6)

TGDS = Thai Geriatric Depression Score, EQ VAS = EQ visual analogue, LDL= low-density lipoprotein, HDL= high-density lipoprotein, DR = diabetic retinopathy, DN = diabetic nephropathy, BMI = Body Mass Index.

**Table 2 healthcare-10-01990-t002:** Model-fitting result of the prospective analysis of depression as a predictor to glycemic control, controlling for covariates (fixed effects) and adjusted for within-person change over time (random effect), number of observations = 577.

	HbA1c					
		Estimate	Std. Error	95% CI		Wald’s *p*
				upper	lower	
Intercept		8.81	0.97	6.95	10.59	<0.001
TGDS		−0.15	0.06	−0.26	−0.03	0.01
TGDS quadratic polynomial		0.015	0.006	0.003	0.03	0.01
Age		−0.02	0.01	−0.05	0.005	0.13
Female		0.22	0.23	−0.20	0.66	0.31
Duration of DM		0.05	0.01	0.02	0.07	<0.001
EQ VAS		−0.01	0.003	−0.02	−0.005	<0.001
LDL cholesterol		0.003	0.003	0.0001	0.006	0.04
Insulin used		0.36	0.14	0.08	0.68	0.01
Risk of malnutrition		0.17	0.16	−0.14	0.48	0.29
Level of physical activity	Low	-	-	-	-	-
	Moderate	0.05	0.10	−0.14	0.24	0.59
	Vigorous	−0.06	0.13	−0.30	0.20	0.65
Cigarette smoke	Never	-	-	-	-	-
	Former	0.22	0.26	−0.27	0.70	0.24
	Current	−0.35	0.27	−0.86	0.17	0.18
Present of DR		−0.11	0.17	−0.43	0.23	0.54
Present of DN		0.19	0.13	−0.06	0.43	0.13
BMI > 25		−0.10	0.14	−0.37	0.16	0.46
Random effects		Estimate	Std.Dev.			
Intercept		0.65	0.80			
Residual		0.67	0.82			

TGDS = Thai Geriatric Depression Score, EQ VAS = EQ visual analogue, LDL= low-density lipoprotein, DR = diabetic retinopathy, DN = diabetic nephropathy, BMI = Body Mass Index, Std.Dev. = standard deviation.

## Data Availability

The datasets used and/or analyzed during the current study are available from the corresponding author on reasonable request.

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
