# Peer review of "The Geriatric Depression Scale Predicts Glycemic Control in Older Adult with Type 2 Diabetes Mellitus: A Longitudinal Study"

_healthcare, 2022, doi:10.3390/healthcare10101990_

Round 1

Reviewer 1 Report

This authors aimed to investigate a longitudinal association between depression, and HbA1c among the elderly diabetes patients. Type 2 DM patients with 60 years and above and normal cognition were recruited from the outpatient department during 1stJune 2020 to 1st July 2021. Thai Geriatric Depression Scale (TGDS) and HbA1c were assessed at five times point (baseline and every 12-week) for 1 year. A linear mixed effect model with random effect was used. The factors associated with HbA1c included demographic, type 2 DM-related and cardiovascular character-istics were recorded at baseline then entered in the model as fixed effects variables. At baseline 14% of 146 patients had depression (TGDS score 6 and above) and there was a significant corre-lation between HbA1c and depression (r = 0.26). In which other factors fixed, the longitudinal analysis indicated that TGDS was a significant predictor of HbA1c and a relationship was J-shaped. An increase of 1 point of TGDS among 5 to 13, corresponded to an increase of HbA1c in the next OPD visit. This data suggests that with the TGDS of 6 and above (susceptibility for de-pression) there is a relationship between depression and subsequent increase in HbA1c in DM pa-tient age 60 and above.

The paper is interesting.

I have some minor suggestions for the authros:

1 The abstract must better summarize the sections

2  Insert a clear purpose

3 Par 2.4.3 is very strange it seems composed with subparagraphs with only titles (2.4.3.1  ,  2.4.2.1, etc…)

4. Par. 2.5 is lacking

5 Check the resolution of the figures

Author Response

Point 1: The abstract must better summarize the sections

Response 1:

Author Response: Thanks very much for your suggestion. We revise the abstract to be more summarize as shown.

Abstract: The present of comorbid depression and diabetes are associated with worse glycemic control, higher complication and mortality risk than expected by each condition alone. The association between various severity of depressive symptoms and glycemic control over time among ype 2 diabetic older patients was unclear. This study aimed to investigate a longitudinal association between depression, and HbA1c among type 2 diabetic older patients. Type 2 diabetes patients with 60 years and above with normal cognition were recruited from the outpatient department during 1stJune 2020 to 1st July 2021. Thai Geriatric Depression Scale (TGDS) and HbA1c were assessed at five times point (baseline and every 12-week) for 1 year. A linear mixed effect model was used. Of 161 enrolled participants, 146 completed the study. At baseline 14% were suscepti-ble to depression or having depression (TGDS score 6 and above) and there was a significant correlation between HbA1c and depression (r = 0.26, p = < 0.01). The longitudinal analysis indi-cated that TGDS was a significant predictor of HbA1c in the next visit and a relationship was J-shaped. TGDS below 5 was associated with decreasing HbA1c in the next visit but the associa-tion turned to positive at TGDS score at 5 or higher. The present of significant symptom of de-pression association with glycemic control in the next 3-month interval OPD visit event though major depressive disorder has not yet been established.

 Point 2: Insert a clear purpose

Response 2:

Author Response: Thanks very much for your suggestion. We have added details about gap and our purpose as shown in the abstract and introduction as shown.

Line 55

Data from Thai National Health Examination Survey 2004–2014 has been shown that prevalence of DM significantly increased with age and peaked at 60 to 69 years old (9), meanwhile normal aging is associated with a progressive increase in HbA1C (10). Specifically, depression effects approximately 17 to 80 % of the Thai older adults in the community depending on the setting and measurement (11-13).

Of the few longitudinal studies, only one by Richardson et.al studied in Type 2 older diabetic patients and demonstrated that depression is associated with persistent-ly higher HbA1c levels (14). Consistent with another longitudinal studies by Lustman et.al and Ching-Ju Chiu et.al in middle-aged and older type 2 diabetic patients (4, 15). Whereas Aikens et al. and Fisher et.al indicated that depression did not predict change in HbA1c levels in prospective analysis (16, 17).

Older adults are focused on the treatment of medical conditions and are not ade-quately recognized the symptoms of depression which mostly less-clear-cut of symp-tom in primary care setting than those seen in psychiatry clinic. Prior studies of the as-sociation between depression and glycemic control were mostly cross-sectional in na-ture. Meanwhile previous longitudinal studies provided insufficient information of depression as a predictor of glycemic control among older adults in primary care set-ting. Therefore, the main purpose of this study is to demonstrate the association be-tween depression and glycemic control in type 2 diabetic older patients. Based upon available evidence in the literature, the hypothesis is that depression would be associ-ated with higher HbA1c levels in the next visit (HbA1c next visit) in the type 2 diabetic older patients. This association will provide the proper care plan of depression screen-ing in primary care setting.

Point 3: Par 2.4.3 is very strange it seems composed with subparagraphs with only titles (2.4.3.1  ,  2.4.2.1, etc…)

Response 3:

Author Response: Thanks very much for your suggestion. We have chang as your suggession as shown.

Line 92

2.2 Study instrruments

We record the demographic characteristics (age, gender); type 2 diabetes-related variables including years of being DM, medications for type 2 DM (no insulin vs. using insulin), present of diabetic retinopathy and present of diabetic nephropathy; and cardiovascular risk factors consisted of body mass index (BMI), cigarette smoking and LDL-cholesterol.

Level of physical activity in the past seven days was categorized as low, moderate and vigorous intensity using the short form of The International Physical Activity Questionnaires (IPAQ)-Thai version which appeared to have acceptable validity and reliability (Spearman’s correlation 0.22 and intraclass correlation coefficient 0.69) (18). The IPAQ short form asks about three specific intensity of activities undertaken in domains including leisure time, domestic and gardening activities, work-related and transport-related activity;

Nutritional status was assessed using the Mini Nutritional Assessment (MNA) scale in Thai version. MNS is a screening for undernutrition in geriatric practice(19). Participants were interviewed by the examiners, the score 12 to 14 indicates “normal nutritional status”, 8 to 11 indicates “at risk of malnutrition”, and less than 8 indicates “malnourished”.

Quality of life was determined using the 5-level EQ-5D  (EQ-5D-5L) Thai version(20) which consists of 2 parts, the EQ-5D descriptive system and the EQ visual analogue scale (EQ VAS). The EQ VAS records the subject’s self-rated health status, where the endpoints of the visual analogue scale are labelled ‘The best health you can imagine’ and ‘The worst health you can imagine’. The score of 100 indicating the best health status.

Depressive symptoms were assessed by Thai Geriatric Depression Scale -15 items (TGDS). It was developed by Wongpakaran N et.al in 2012 and found well perform at the geriatric outpatients care with the cut-off score of > 5 a sensitivity is 0.92 and a specificity is 0.87 (21). The questionnaire can be self-administered or presented as an interview. Most of the participants in this study answered the 15 questions in the yes/no format themselves. For the older adults with eye problems and low literature, the trained examiner read all items to them in a clear monotone voice without facial expression. Each item scored one point if positively answered except for item 1, 5, 7, 11, 15 which received points when negatively answered. Sum scores of 0-5 are considered normal, 6-10 indicated susceptible to depression, and 11-15 indicated that the patient has depression.

Hemoglobin A1C (HbA1C), serum cholesterol, HDL-cholesterol, triglycerides and LDL-cholesterol were measured at every visit. Blood samples were drawn after an overnight fast (at least 8 hours fasting).

Thereafter, participants received routine diabetes care in OPD, which pro-vided by physician following detailed treatment algorithms base on “Treatment of Diabetes in Older Adults: An Endocrine Society Clinical Practice Guideline”(22) to maintain their established oral hypoglycemic and/or insulin therapy.

Point 4: Par. 2.5 is lacking

Response 4:

Author Response: Thanks very much for your suggestion. We have corrected it your suggestion.

Point 5: Check the resolution of the figures

Response 5:

Author Response: Thanks very much for your suggestion. We have corrected it as shown in the manuscript.

Reviewer 2 Report

Dear Authors, 

Thank you for a possibility to review your work. Overall, it's an interesting manuscript - its' definite strenght is a prospective design. I do have a few remarks I would like to share with you.

General - I suggest changing the term "elderly" into "older" or "older adults" as more neutral (not peyorative in nature).

Intro - it is not clear to me what you paper adds to existing research - it seems rather replicative - in what way is it new? I would also expect more review regarding covariates included in the model, so that it becomes clear why you decided to investigate these specific ones and not the others (such as self assessed health for example). 

Line 56 - could you describe more directly what the gaps are?

Study design and participatns - I suggest moving into this section informatin regarding basic characteristics of your participants from the Results.  How they were approached? What was the response rate?  Could you describe the procedure itself a bit more (i.e. were all patients able to fill the questionnaires by themselves? Were they getting any assistance?)

2.2. - line 83 - could you please indicate in what way the tool performs better? How it should be understood? 

2.4. - I suggest to reorganize this section. It seem to combine description of methods and of the model. The latter could suit better into Statistical methods sections. Then, not so many headings would be needed.

Analyses - Could you explain why did you assume non linear relationships? Did you check for interactions in the model? 

Discussion - How would you explain a J-shaped relationship? Why lower levels of depression result in worse glycemic control? 

The possible impact of covariates seem to be neglected in the disucssion as well as in the limitations.

Author Response

Response to Reviewer 2 Comments

Point 1: I suggest changing the term "elderly" into "older" or "older adults" as more neutral (not peyorative in nature).
Response 1: 
Author Response: Thanks very much for your suggestion. We have change from “elderly” to older adult as your suggestion, for the example.
    … The goal of this study is to demonstrate the association between TGDS, a simple self-reported depressive symptoms questionnaire, and HbA1c next visit. This result highlights the importance of comprehensive care in the type 2 diabetic older patients to not only focus at target HbA1c or DM complication but also con-sider screening for depression as a part of DM care. The older adult with depressive symptoms or disorder needs ongoing monitoring of depression recurrence within the context of routine care…...

Point 2: Intro - it is not clear to me what you paper adds to existing research - it seems rather replicative - in what way is it new? I would also expect more review regarding covariates included in the model, so that it becomes clear why you decided to investigate these specific ones and not the others (such as self assessed health for example).
Response 2: Response 2: 
Author Response: Thanks very much for your suggestion. We have added details about gap and our purpose as shown in the abstract and introduction as shown.

Abstract: The present of comorbid depression and diabetes are associated with worse glycemic control, higher complication and mortality risk than expected by each condition alone. The association between various severity of depressive symptoms and glycemic control over time among type 2 diabetic older patients was unclear. This study aimed to investigate a longitudinal association between depression, and HbA1c among type 2 diabetic older patients. Type 2 diabetes patients with 60 years and above with normal cognition were recruited from the outpatient department during 1stJune 2020 to 1st July 2021. Thai Geriatric Depression Scale (TGDS) and HbA1c were assessed at five times point (baseline and every 12-week) for 1 year. A linear mixed effect model was used. Of 161 enrolled participants, 146 completed the study. At baseline 14% were susceptible to depression or having depression (TGDS score 6 and above) and there was a significant correlation between HbA1c and depression (r = 0.26, p = < 0.01). The longitudinal analysis indicated that TGDS was a significant predictor of HbA1c in the next visit and a relationship was J-shaped. TGDS below 5 was associated with decreasing HbA1c in the next visit but the association turned to positive at TGDS score at 5 or higher. The present of significant symptom of depression association with glycemic control in the next 3-month interval OPD visit event though major depressive disorder has not yet been established.

Line 55
Data from Thai National Health Examination Survey 2004–2014 has been shown that prevalence of DM significantly increased with age and peaked at 60 to 69 years old (9), meanwhile normal aging is associated with a progressive increase in HbA1C (10). Specifically, depression effects approximately 17 to 80 % of the Thai older adults in the community depending on the setting and measurement (11-13). 
Of the few longitudinal studies, only one by Richardson et.al studied in Type 2 older diabetic patients and demonstrated that depression is associated with persistent-ly higher HbA1c levels (14). Consistent with another longitudinal studies by Lustman et.al and Ching-Ju Chiu et.al in middle-aged and older type 2 diabetic patients (4, 15). Whereas Aikens et al. and Fisher et.al indicated that depression did not predict change in HbA1c levels in prospective analysis (16, 17). 
Older adults are focused on the treatment of medical conditions and are not adequately recognized the symptoms of depression which mostly less-clear-cut of symptom in primary care setting than those seen in psychiatry clinic. Prior studies of the association between depression and glycemic control were mostly cross-sectional in nature. Meanwhile previous longitudinal studies provided insufficient information of depression as a predictor of glycemic control among older adults in primary care set-ting. Therefore, the main purpose of this study is to demonstrate the association be-tween depression and glycemic control in type 2 diabetic older patients. Based upon available evidence in the literature, the hypothesis is that depression would be associated with higher HbA1c levels in the next visit (HbA1c next visit) in the type 2 diabetic older patients. This association will provide the proper care plan of depression screening in primary care setting.

Point 3: Line 56 - could you describe more directly what the gaps are?
Response 3: Author Response: Thanks very much for your suggestion. The gaps are revise more clearly in the introduction as shown.

Line 66
Older adults are focused on the treatment of medical conditions and are not adequately recognized the symptoms of depression which mostly less-clear-cut of symptom in primary care setting than those seen in psychiatry clinic. Prior studies of the association between depression and glycemic control were mostly cross-sectional in nature. Meanwhile previous longitudinal studies provided insufficient information of depression as a predictor of glycemic control among older adults in primary care set-ting. Therefore, the main purpose of this study is to demonstrate the association be-tween depression and glycemic control in type 2 diabetic older patients. Based upon available evidence in the literature, the hypothesis is that depression would be associated with higher HbA1c levels in the next visit (HbA1c next visit) in the type 2 diabetic older patients. This association will provide the proper care plan of depression screening in primary care setting.

Point 4: Study design and participatns - I suggest moving into this section information regarding basic characteristics of your participants from the Results.  How they were approached? What was the response rate?  Could you describe the procedure itself a bit more (i.e. were all patients able to fill the questionnaires by themselves? Were they getting any assistance?)

Response 4: Author Response: Thanks very much for your suggestion. We the revise the topic Study design and participants as shown.

Line 80
2.1. Study Design and Participants

This is a longitudinal study that aimed to analyze the relationship of depression with subsequent glycemic control over one year follow up period in which the variables were repeated measure at 3-month duration for 5 times. Participants were recruited from older patients (age 60 years and above) diagnosed with type 2 diabetes mellitus and ongoing treatment at outpatient department (OPD) of a primary care setting that provided service by university-affiliated physicians in Southern Thailand. The type 2 diabetic older patients who came for routine visit during the period of study on the 1st June 2020 to the 1st July 2021 received the invitation poster that was approved by Ethical Committee and voluntary applied for participation.  
Participants with an ICD-10 code for dementia, cognitive impairment, depressive disorder or anxiety, bipolar disorder, psychotic disorder, alcoholism or drug addiction within 1 year prior the first visit, malignancy within 5 years prior the first visit were excluded. Cognitive function and risk of suicide were evaluated at baseline using Thai Mini - Mental State Examination (MMSE-Thai 2002) and 8-questions assessment of suicide risk tool (8Q) respectively to exclude participants with cognitive impairment, active suicidal ideation from the study. The high suicidal risk subjects (8Q score more than 17) will be sent to psychiatric clinic.

Among 161 participants, 146 completed 5-time visit (Time0 to Time4) during one year of longitudinal study. Withdrew from the study were 8 participants who declined hospital visit due to COVID 19 pandemic and 7 participants were lost to follow-up.
Majority of the participants were independent and socially active, meanwhile 8 participants (5.48%) need assistance in answering the questionnaires.

Point 5: 2.2. - line 83 - could you please indicate in what way the tool performs better? How it should be understood?
Response 5: Author Response: Thanks very much for your suggestion. We the revise the topic Study instrument as shown.

Line 124
Depressive symptoms were assessed by Thai Geriatric Depression Scale -15 items (TGDS). It was translated by Wongpakaran et.al in 2012  and found well perform at the geriatric outpatients care with the cut-off score of > 5 a sensitivity is 0.92 and a specificity is 0.87 (21). The Geriatric Depression Scale focuses specifically on psychiatric rather than somatic symptoms (sleep disturbances, weight loss and pessimism about the future) which can be related to aging than the younger adults. The questionnaire can be self-administered or presented as an interview. Most of the participants in this study answered the 15 questions in the yes/no format themselves. For the older adults with eye problems and low literature, the trained examiner read all items to them in a clear monotone voice without facial expression. Each item scored one point if positively answered except for item 1, 5, 7, 11, 15 which received points when negatively answered. Sum scores of 0-5 are considered normal, 6-10 indicated susceptible to depression, and 11-15 indicated that the patient has depression. 

Point 6: I suggest to reorganize this section. It seem to combine description of methods and of the model. The latter could suit better into Statistical methods sections. Then, not so many headings would be needed.
Response 6: Author Response: Thanks very much for your suggestion. We the revise the topic Study instrument, Sample Size Estimation and Statistical Analysis as shown.

Line 101-166.

2.2. Study instruments
We record the demographic characteristics (age, gender); type 2 diabetes-related variables including years of being DM, medications for type 2 DM (no insulin vs. using insulin), present of diabetic retinopathy and present of diabetic nephropathy; and car-diovascular risk factors consisted of body mass index (BMI), cigarette smoking and LDL-cholesterol. 
Level of physical activity in the past seven days was categorized as low, moderate and vigorous intensity using the short form of The International Physical Activity Questionnaires (IPAQ)-Thai version which appeared to have acceptable validity and reliability (Spearman’s correlation 0.22 and intraclass correlation coefficient 0.69) (18). The IPAQ short form asks about three specific intensity of activities undertaken in domains including leisure time, domestic and gardening activities, work-related and transport-related activity;
Nutritional status was assessed using the Mini Nutritional Assessment (MNA) scale in Thai version. MNS is a screening for undernutrition in geriatric practice(19). Participants were interviewed by the examiners, the score 12 to 14 indicates “normal nutritional status”, 8 to 11 indicates “at risk of malnutrition”, and less than 8 indicates “malnourished”. 

Quality of life was determined using the 5-level EQ-5D (EQ-5D-5L) Thai ver-sion(20) which consists of 2 parts, the EQ-5D descriptive system and the EQ visual analogue scale (EQ VAS). The EQ VAS records the subject’s self-rated health status, where the endpoints of the visual analogue scale are labelled ‘The best health you can imagine’ and ‘The worst health you can imagine’. The score of 100 indicating the best health status.

Depressive symptoms were assessed by Thai Geriatric Depression Scale -15 items (TGDS). It was translated by Wongpakaran et.al in 2012  and found well perform at the geriatric outpatients care with the cut-off score of > 5 a sensitivity is 0.92 and a specific-ity is 0.87 (21). The Geriatric Depression Scale focuses specifically on psychiatric rather than somatic symptoms (sleep disturbances, weight loss and pessimism about the fu-ture) which can be related to aging than the younger adults. The questionnaire can be self-administered or presented as an interview. Most of the participants in this study answered the 15 questions in the yes/no format themselves. For the older adults with eye problems and low literature, the trained examiner read all items to them in a clear monotone voice without facial expression. Each item scored one point if positively answered except for item 1, 5, 7, 11, 15 which received points when negatively answered. Sum scores of 0-5 are considered normal, 6-10 indicated susceptible to depression, and 11-15 indicated that the patient has depression. 

Hemoglobin A1C (HbA1C), serum cholesterol, HDL-cholesterol, triglycerides and LDL-cholesterol were measured at every visit. Blood samples were drawn after an overnight fast (at least 8 hours fasting). 

Thereafter, participants received routine diabetes care in OPD, which pro-vided by physician following detailed treatment algorithms base on “Treatment of Diabetes in Older Adults: An Endocrine Society Clinical Practice Guideline”(22) to maintain their established oral hypoglycemic and/or insulin therapy.

2.3 Sample Size Estimation
The sample size calculations for longitudinal data was done using longpower package, R program (23). The authors assume weak correlation coefficient (rho) in the range of – 0.3 < rho < 0.3 and the large error variance (sigma2). The sample size when rho = 0.2 and sigma2 = 300 is 132.

2.4. Statistical Analysis
Descriptive results of the participants in this study are express as the calculation of means and standard deviation (SD) for continuous variables and as frequencies and percentages for categorical variables. This longitudinal study aimed to analyze the rela-tionship of depression with subsequent HbA1c next visit in which the variables were repeated measure at 12-week duration for 5 times. 
To perform prospective analyses, a linear mixed effect model with random effect to adjust for variance within individual and variation between individuals over time were implemented. There were two levels of variance components. Level 1 considered within person change over time and level 2 considered between individuals’ variation regardless of time. Time-variant covariates including HbA1c, TGDS, Quality of life, BMI, diabetic complications, medication, nutritional status, cigarette smoking, quality of life and physical activity were collected at each time point from T0 to T4 in a 12- week duration As HbA1c is the test that reflects average plasma glucose of the 12 weeks previously so we consider that the individual differences on current HbA1c measured (time n) are related to individual differences on other variables measured at a previous occasion (time n-1) as shown in Figure 1.
Data was analyzed using R program with 2-sided tests of hypotheses. A p-value <0.05 on the two-tail was considered to indicate statistical significance

Point 7: Analyses - Could you explain why did you assume non linear relationships? Did you check for interactions in the model?
Response 7: Author Response: Thanks very much for your suggestion.
This is the explanation on why we assumed non-linear relationship in this study.

Crude effect and adjusted effect of the estimation of each variable in the last model was check 
Figure 1 Show Adjusted estimate of the model

The estimation of crud effect and adjusted effect did not difference more than 20%.

For the example

For the example

                       Crude Estimate     Adjusted Estimate

dmduration     0.05296                0.046408

LOESS Curve Fitting (Local Polynomial Regression) was used for fitting a smooth curve between two variables, TGDS and HbA1c. With the polynomial degree = 2, the estimated curve fits (blue line) was separated from the population curve which shown in red line as shown in figure 2. So the author hypothesize that the relationship between TGDS and HbA1c is not linear so the author analyzed TGDS as a quadratic polynomial. 

Figure 2 Local Polynomial Regression to estimate the relationship between TGDS as a predictor to glycemic control. The blue line represents a non-parametric loess fit and the red line represents a linear fit.

***Figure 1 and 2 cannot upload hear please review on the uploaded file*** 

Point 8: Discussion - How would you explain a J-shaped relationship? Why lower levels of depression result in worse glycemic control?
Response 8: Author Response: Thanks very much for your suggestion. We the revise the topic Discussion as shown.
Line 238
GDS is one of the screening tools that has been used in many studies and been translated and validated into many languages including Thai version called TGDS by Wongpakaran N et.al (21). As the best of our reviewed, this is the first study that re-port a non-linear “J-shaped” relationship of GDS with HbA1c.

Similar associations were observed for the other depressive screening tools. The previous longitudinal study by Abuhegzy et.at in 2017 found a significant association using Beck Depression Inventory score with hyperglycemia (27). Study of the association between cardiovascular risk factors and concurrent depressive symptoms using hospital anxiety and depression score (HADS-D) in patients with three common cardiometabolic conditions by Bhautesh et.at in 2014 demonstrated the J-shape association of HbA1C with probability of having a positive HADS-D (>7). Not only HbA1c but sys-tolic blood pressure, diastolic blood pressure, BMI and total cholesterol also demon-strated the J-shape association with probability of having a positive HADS-D(27). Studies also demonstrated a non-linear “J-shaped” relationship between GDS and var-ies health variables. The Health in Men Study found the “J-shaped” association be-tween the numbers of years lived with the diagnosis of DM with increase the odds of current depression (28).

Interestingly, TGDS less than 6 had negative association with HbA1c. The expla-nation should be one of the limitation of our study that we did not evaluate other men-tal health conditions that can alter HbA1c or the relationship between depression and HbA1c included stress, other mood disorders such as bipolar disorder and diabetes distress. Stress that lasts for many weeks or months can lead to unstable levels of blood glucose. The fluctuations of blood glucose, whether high or low, may cause a person to experience a variety of symptoms such as hunger, confusion, aggression and irritability. Stress is a normal part of daily life, people with stress can live without depression but may lead to unhealthy eating behavior that effect glycemic control.    

Point 9: The possible impact of covariates seem to be neglected in the disucssion as well as in the limitations.
Response 9: Response 8: Author Response: Thanks very much for your suggestion. We the revise the topic Discussion as shown.
Line 254
Interestingly, TGDS less than 6 had negative association with HbA1c. The expla-nation should be one of the limitation of our study that we did not evaluate other men-tal health conditions that can alter HbA1c or the relationship between depression and HbA1c included stress, other mood disorders such as bipolar disorder and diabetes distress. Stress that lasts for many weeks or months can lead to unstable levels of blood glucose. The fluctuations of blood glucose, whether high or low, may cause a person to experience a variety of symptoms such as hunger, confusion, aggression and irritability. Stress is a normal part of daily life, people with stress can live without de-pression but may lead to unhealthy eating behavior that effect glycemic control.    
